# Multiple Injections of Platelet-Rich Plasma Versus Hyaluronic Acid for Knee Osteoarthritis: A Systematic Review and Meta-Analysis of Current Evidence in Randomized Controlled Trials

**DOI:** 10.3390/jpm13030429

**Published:** 2023-02-27

**Authors:** Shang Li, Fei Xing, Tongtong Yan, Siya Zhang, Fengchao Chen

**Affiliations:** 1Medical Cosmetic Center, Beijing Friendship Hospital, Capital Medical University, Beijing 100050, China; 2Department of Orthopedics, Orthopedic Research Institute, West China Hospital, Sichuan University, No. 37 Guoxue Lane, Chengdu 610041, China

**Keywords:** platelet-rich plasma, knee osteoarthritis (KOA), hyaluronic acid, pain

## Abstract

In recent years, various clinical trials have focused on treating knee osteoarthritis (KOA) with multiple injections of platelet-rich plasma (PRP). However, compared with the multiple hyaluronic acid (m-HA) injections, the clinical efficacy of multiple PRP (m-PRP) injections for KOA still remains controversial among these studies. Therefore, we aimed to compare the clinical effectiveness of m-PRP injections with m-HA injections in the treatment of KOA in this systematic review. Relevant clinical trials were searched via bibliographic databases, including Medline, PubMed, Embase, Web of Science, and Cochrane Central Register of Controlled Trials, to compare the m-PRP and m-HA injections in the treatment of KOA. Finally, fourteen randomized controlled trials, including 1512 patients, showed the postoperative VAS, WOMAC, IKDC, or EQ-VAS scores and were enrolled in this systematic review. Compared with the group of intra-articular m-HA injections, the group of intra-articular m-PRP injections was lower in the VAS scores at 3-month (WMD = −0.25; 95%CI, −0.40 to −0.10, *p* = 0.0009) and 12-month (WMD = −0.64; 95%CI, −0.79 to −0.49, *p* < 0.00001) follow-ups. In addition, the group of intra-articular m-PRP injections was also lower in the WOMAC scores at 1-month (WMD = −1.23; 95%CI, −2.17 to −0.29, *p* = 0.01), 3-month (WMD = −5.34; 95%CI, −10.41 to −0.27, *p* = 0.04), 6-month (WMD = −11.02; 95%CI, −18.09 to −3.95, *p* = 0.002), and 12-month (WMD = −7.69; 95%CI, −12.86 to −2.52, *p* = 0.004) follow-ups. Furthermore, compared with the group of intra-articular m-HA injections, the group of intra-articular m-PRP injections was higher in the IKDC scores at 3-month (WMD = 7.45; 95%CI, 2.50 to 12.40, *p* = 0.003) and 6-month (WMD = 5.06; 95%CI, 1.94 to 8.18, *p* = 0.001) follow-ups. However, the long-term adverse side of m-PRP injections for KOA still needs more large-scale trials and long-term follow-ups.

## 1. Introduction

Knee osteoarthritis (KOA), as a common degenerative disease, results in serious musculoskeletal disorders such as constant pain, stiffness, swelling, and knee dysfunction [1,2]. Over the past few years, the incidence of KOA has constantly increased around the world, which has introduced heavy burdens on healthcare systems worldwide. Because of the recurring symptoms and repetitive treatments, most patients with KOA live with a low quality of life, which also introduces a heavy financial burden on the KOA patients’ families [3]. Currently, various therapeutic methods have been used in treating KOA, including patient education, exercise therapy, pharmacotherapy [4], physical therapy, and joint replacement [5]. In addition, more and more researchers have focused on developing minimally invasive interventions for KOA, which can effectively control the symptomatic progression [6]. Among these minimally invasive interventions, intra-articular (IA) injections are safe and effective, and are commonly used in clinical treatment to control the symptomatic progression of KOA [7].

Hyaluronic acid (HA) is a type of commonly used agent for IA injections, which is a natural glycosaminoglycan in the articular cavity [8]. In addition, HA could regulate the cellular microenvironment and contributes to improving the viscoelastic characteristic of synovial fluid in the articular cavity [9]. Furthermore, intra-articular HA injections can increase synovial fluid volumes, which is beneficial for restoring knee functions in KOA patients. After being injected into the articular cavity, HA was degraded into lower molecular weight products. Recently, the effectiveness of HA for KOA has been confirmed by several clinical studies [10,11]. In addition, HA could relieve pain and restore knee function. Furthermore, the previous study demonstrated that two or more injections could increase the effectiveness of HA in the treatment of KOA [12].

Platelet-rich plasma (PRP) is a type of autologous biological product extracted from whole blood and that contains various growth factors [13], such as vascular endothelial growth factor (VEGF), fibroblast growth factor (FGF), and platelet-derived growth factor (PDGF) [14,15,16]. Over the past few years, PRP attracted more attention in treating KOA patients because of its potential therapeutic value in repairing cartilage [17,18]. It has been found that PRP was able to increase the proliferative capacity of chondrocytes, modulate the microenvironment, and reduce inflammatory reactions [17]. Many studies confirmed that PRP could relieve pain and improve knee function. In addition, multiple injections of PRP are more effective than a single PRP injection [19,20].

Recently, several clinical studies focused on multiple PRP (m-PRP) injections versus multiple HA (m-HA) injections in treating KOA [21,22]. However, the efficacy and safety of m-PRP and m-HA injections remain controversial in these studies, and there has been no related meta-analysis published yet. Therefore, we compared the clinical effectiveness of m-PRP and m-HA injections in KOA therapy and analyzed the results in this systematic review.

## 2. Materials and Methods

The study was conducted following the Preferred Reporting Items for Systematic reviews and Meta-Analysis (PRISMA) guidelines [23].

### 2.1. Literature Search

The related studies were independently acquired by two reviewers through electronic databases, comprising Cochrane Central Register of Controlled Trials (December 2022), MEDLINE, PubMed (1966 to December 2022), Web of Science (1990 to December 2022), and Embase (1974 to December 2022). The Google search engine (December 2022) was also used to search for additional eligible studies. The electronic search strategies were as follows: “platelet-rich plasma”, “PRP”, “autologous plasma”, “hyaluronic acid”, “HA”, “osteoarthritis”, “knee osteoarthritis”, “OA”, “KOA”, and “multiple”. In all included electronic databases, a strategy was used indifferently when conducting searches. Studies in human bodies focused on treating KOA, and multiple IA injections of PRP or HA were searched. And the unpublished studies were researched from international register of clinical trials, ClinicalTrials.gov. We tried to email the authors if we found related uncomplete RCTs to acquire the data. In addition, we also utilized the method of backward chaining references from retrieved papers to find relevant studies in retrieved papers and to maximize the search.

### 2.2. Eligibility Criteria

The exclusion criteria were the following: (1) animal studies; (2) editorial, poster, experimental studies, cohort, and observational studies, and cadaveric and biomechanics studies; (3) publishing language was not English; (4) protocol descriptions and technical notes; (5) duplicated publications; (6) systematic reviews and meta-analyses; (7) single abstracts, comment papers, case reports, and correspondence; (8) the participants were involved in recent and/or imminent knee surgery; (9) no outcome interest. When two reviewers disagreed about the inclusion and exclusion criteria, the disagreement was solved via consultation or by a third reviewer.

The inclusion criteria were performed as follows: (1) The studies were randomized controlled trials (RCTs). (2) Studies focused on the outcomes of IA injections for KOA. (3) Studies involved the administration of multiple IA PRP injections. (4) The control group should be treated with multiple IA HA injections. (5) Unpublished studies—that were relevant studies not in the databases—were also included. (6) Only articles in English were examined. (7) The studies described the procedures of PRP injections, injection frequency, and PRP dosage performed on participants. 

The exclusion criteria included the following: (1) animal studies; (2) editorial, poster, experimental studies, cohort, and observational studies, and cadaveric and biomechanics studies; (3) publishing language was not English; (4) protocol description and technical notes; (5) duplicated publications; (6) systematic reviews and meta-analysis; (7) single abstracts, comment papers, case reports, and correspondence; (8) the participants were involved recent and/or imminent knee surgery; (9) no outcome interest. When two reviewers disagreed about the inclusion and exclusion criteria, the disagreement is solved via consultation or a third reviewer.

### 2.3. Data Extraction

In all included studies, data extraction was performed by two reviewers independently. The demographic characteristics, including the first author, year of publication, sample size, average age of participants, male ratio, body mass index (BMI), symptom duration, and follow-ups, were extracted for this systematic review. The interventional factors, including the procedures of PRP, excluding platelets count, HA component, PRP dosage, and HA dosage, were extracted in this study. If there were disputes during extractions, they were resolved by discussion and consensus with a third reviewer. Apart from that, descriptive statistics were performed, and parameters were analyzed in each study by two reviewers.

### 2.4. Outcome Measures

We evaluated the clinical efficacy and safety of m-PRP and m-HA injections in patients with KOA. The outcomes comprise a visual analog scale (VAS), the Western Ontario and McMaster Universities Arthritis Index (WOMAC), the International Knee Documentation Committee (IKDC), and EuroQol visual analog scale (EQ-VAS) scores.

### 2.5. Assessment of Methodological Quality

The methodological quality of enrolled RCTs was assessed by two reviewers independently. The bias of all RCTs was evaluated using Modified Jadad scores in this study [24]. If the modified Jadad scores were ≥4 points, the RCTs were considered to be of high quality.

### 2.6. Statistical Analysis

The statistical analysis was conducted using RevMan Manager 5.3 (The Cochrane Collaboration, Oxford, UK) by two reviewers independently. *p* < 0.05 was considered statistically significant. For continuous variables, such as WOMAC scores, weight mean differences (WMDs) were estimated with a 95% confidence interval (95%CI). Statistical heterogeneity for enrolled trials was assessed via Q chi-square test and the I^2^ statistic. Moreover, heterogeneity was reported as high, and the randomized-effects model was performed when I^2^ > 50%. Then, the fixed-effect model was chosen. For all enrolled studies, the different outcomes of m-PRP and m-HA injections were presented using forest plots.

## 3. Results

### 3.1. Study Selection

Finally, a total of 55 related studies were retrieved from databases using a search of the literature, and no unpublished studies were retrieved from the registration website. In total, 32 studies of them were duplicates and excluded. After that, the titles and abstracts of 23 literature studies were assessed, and 7 were excluded after the assessment, for they did not meet the selection criteria. Finally, 14 RCTs [22,25,26,27,28,29,30,31,32,33,34,35,36,37], comprising 1512 participants and published between 2012 and 2022, matched the selection criteria and were enrolled in this meta-analysis. The flow chart of the literature research is shown in Figure 1.

### 3.2. Study Characteristics

The demographic characteristics of all the enrolled studies are presented in Table 1. In total, 14 studies, including 1512 patients, described the administration of m-PRP injections in treating KOA. The sex ratio of all included participants was 0.8 (M/F). In total, 781 out of 1512 (51.65%) patients were treated with multiple doses of IA PRP injections, and 44.81% were male. In total, 731 out of 1512 (48.35%) patients were treated with multiple doses of IA HA injections, and 44.05% were male. The mean age ranged from 46.2 to 66.5 years, the mean BMI ranged from 22.5 ± 2.3 kg/m^2^ to 28.47 ± 4.54 kg/m^2^, and the mean symptom duration ranged from 11.5 ± 2.6 months to 9.7 ± 3.9 years. The sample size of the m-PRP group ranged from 25 to 104, and the m-HA group ranged from 28 to 88, the mean age ranged from 51.5 to 66.2 years, the mean BMI ranged from 22.8 ± 2.1 kg/m^2^ to 29.98 ± 5.24 kg/m^2^, and the mean symptom duration ranged from 10.5 ± 2.0 months to 10.1 ± 4.2 years. Among these RCTs, five were conducted in Turkey, five in Italy, two in Iran, and one in Egypt and Serbia respectively. Moreover, the follow-ups of these studies ranged from 6 to 24 months.

The intervention information is presented in Table 2. The frequency of injections included once a week, once every 2 weeks, once every 3 weeks, and once every 4 weeks. In ten studies, the injection interval was the same in both groups, while in four studies, the intervals of m-PRP group were longer than that of the m-HA group. Among these studies, the volume of whole blood was different. Three studies collected 150 mL of whole blood samples, which were subsequently centrifuged twice, producing 20 mL of PRP, divided into four doses during treatments [22,26,32]. Five studies collected 35–60 mL of whole blood, and the blood was centrifuged two times, producing 4–6 mL of PRP, which was one dose [27,30,35,36,37]. Four studies collected 8–20 mL of whole blood, and the blood was centrifuged one time, producing one dose PRP [28,29,33,34]. One study chose PRP from Sigma-Aldrich [31]. One study did not describe the detail of the PRP procedure [25]. Moreover, the dosages of PRP were different, from 2 to 14 mL In contrast, the dosage of HA was 2 mL in all trials. Among these studies, patients in the experimental group received treatments of m-PRP injections only, and the control group received treatments of multiple pure HA injections only.

### 3.3. Risk of Bias

Figure 2 presents the methodological quality of every included study. Additionally, Figure 3 showed the risk of bias in these studies. All of these biases were evaluated by two reviewers independently in this study. The modified Jadad scores of all enrolled RCTs are presented in Table 3. The mean of the modified Jadad scores of all enrolled RCTs was 4.79 (range from 2 to 7), which indicated that most enrolled RCTs were considered high quality.

### 3.4. VAS Scores

In total, six studies, including 463 patients, reported the VAS scores after m-PRP or m-HA injections [28,30,34,35,36,37]. At the 1 month (WMD = 0.03; 95%CI, −0.11 to 0.18, *p* = 0.67) follow-up, no significant differences in VAS score were found between m-PRP and m-HA injections groups. At 3 month (WMD = −0.25; 95%CI, −0.40 to −0.10, *p* = 0.0009) and 12 month (WMD = −0.64; 95%CI, −0.79 to −0.49, *p* < 0.00001) follow-ups, the VAS scores of m-PRP injections were significantly lower than those of m-HA injections (Figure 4). No significant heterogeneities in VAS scores were found at 1 month (*p* = 0.21, I^2^ = 36%), 3 month (*p* = 0.181, I^2^ = 45%), and 12 month (*p* = 0.13, I^2^ = 55%) follow-ups.

### 3.5. WOMAC Scores

In total, 6 studies, including 463 patients, reported the VAS scores after m-PRP or m-HA injections [25,30,31,33,34,35]. The WOMAC scores of m-PRP injections were significantly lower than that of m-HA injections at 1 month (WMD = −1.23; 95%CI, −2.17 to −0.29, *p* = 0.01), 3 month (WMD = −5.34; 95%CI, −10.41 to −0.27, *p* = 0.04), 6 month (WMD = −11.02; 95%CI, −18.09 to −3.95, *p* = 0.002), and 12 month (WMD = −7.69; 95%CI, −12.86 to −2.52, *p* = 0.004) follow-ups. (Figure 5) No significant heterogeneity in WOMAC scores was found at the 1 month (*p* = 0.19, I^2^ = 40%) follow-up. However, there was significant heterogeneity in WOMAC scores at 3 month (*p* < 0.00001, I^2^ = 92%), 6 month (*p* < 0.00001, I^2^ = 96%) and 12 month (*p* < 0.00001, I^2^ = 89%) follow-ups.

### 3.6. IKDC Scores

In total, 6 studies, including 673 patients, reported the IKDC scores after m-PRP or m-HA injections [22,26,28,32,33,35]. The IKDC scores of m-PRP injections were significantly higher than that of m-HA injections at 3 month (WMD = 7.45; 95%CI, 2.50 to 12.40, *p* = 0.003) and 6 month (WMD = 5.06; 95%CI, 1.94 to 8.18, *p* = 0.001) follow-ups. Additionally, no significant differences were found between the groups of m-PRP and m-HA injections at the 12 month (WMD = 3.01; 95%CI, −0.70 to 6.72, *p* = 0.11) follow-up. (Figure 6) No significant heterogeneities in IKDC scores were found at 3 month (*p* = 0.64, I^2^ = 0%), 6 month (*p* = 0.09, I^2^ = 53%), and 12 month (*p* = 0.75, I^2^ = 0%) follow-up.

### 3.7. EQ-VAS Scores

In total, 2 studies, including 350 patients, reported the EQ-VAS scores after m-PRP or m-HA injections [22,32]. No significant differences were found in EQ-VAS scores between groups of m-PRP and m-HA injections at the 6 month (WMD = 2.58; 95%CI, −0.37 to 5.52, *p* = 0.09) follow-up. The EQ-VAS scores of the m-PRP injections were significantly higher than that of m-HA injections at the 12 month (WMD = 2.90; 95%CI, 1.29 to 4.51, *p* = 0.0004) follow-up (Figure 7). Additionally, no significant heterogeneities were found at 6 month (*p* = 0.74, I^2^ = 0%) and 12 month (*p* = 0.75, I^2^ = 0%) follow-ups.

### 3.8. Adverse Effects

In total, 4 studies, including 324 patients, reported complications after m-PRP or m-HA injections [34,35,36,37]. The rate of local pain after injection in the group of m-PRP injections was significantly higher than that of m-HA injections (RD = 0.10; 95%CI, 0.01 to 0.18, *p* = 0.02). There were no significant differences in the rate of local swelling (RD = 0.06; 95%CI, −0.02 to 0.15, *p* = 0.16) and complications (RD = 0.07; 95%CI, −0.05 to 0.20, *p* = 0.24) between m-PRP and m-HA groups (Figure 8). No significant heterogeneities were found in the rate of local pain after injections (*p* = 0.55, I^2^ = 0%) and in the rate of local swelling after injections (*p* = 0.82, I^2^ = 0%). Moreover, there was significant heterogeneity in the rate of complications after injections (*p* = 0.0006, I^2^ = 83%).

### 3.9. Sensitivity Analysis

In sensitivity analysis, each study was removed individually from the overall pooled analysis to assess if the pooled results changed. The results of this meta-analysis are stable.

## 4. Discussion

In this study, we performed this systematic review to compare the clinical effect of m-PRP and m-HA injections in treating KOA. The results demonstrated that m-PRP injections were more effective in relieving pain at 3-month and 12-month follow-ups and could significantly improve knee function—according to MOWAC scores, IKDC scores, and EQ-VAS scores—compared with m-HA injections, making m-PRP a potential method in future research in treating KOA.

Currently, IA injections have been a common therapy in treating KOA due to their several advantages, such as minor wounds, rapid effect, and minimizing systematic complications. Up until now, there have been several therapeutic medicines during IA treatment, such as steroids, HA, PRP, and stem cells. The IA steroid was one of the most common choices, as it could quickly play an anti-inflammatory reaction and relieve pain post-injection [38]. However, steroids would be absorbed into the systematic circulation, resulting in temporary effects and unexpected effects [39]. Thus, the IA steroid was not the best choice for the long-term treatment of KOA. Stem cells are rarely used in large-scale clinical trials because of the ethical principles and immunogenicity. Based on the disadvantages of steroids and stem cells, many clinical trials focused on m-PRP injections in order to search for possibilities for long-term treatments. Among these studies, m-HA injections were used as the control group in order to compare the clinical effectiveness of m-PRP injections in treating KOA.

In the human body, HA is a type of natural ingredient of synovial fluid and articular cartilage [40]. The synovial fluid is important in arthrosis, which contributes to absorbing shock during movement, lubricating cartilage and encouraging the repairment of cartilage and bone. In addition, HA contributes to modulating the inflammation microenvironment of articular cartilage. It was confirmed that the HA concentration of synovial fluids would decrease during KOA progression, which resulted in losing of viscoelastic properties [41]. So that, m-HA injections could restore the dysfunction and decrease KOA progression. HA also presented antioxidative and anti-inflammatory properties in treating KOA—which could decrease the inflammation in articular cartilage and periarticular tissues via reducing local nitric oxide, hydroxyl radicals, and inflammatory relative cytokines—to preserve chondrocyte from programmed cell death and mitochondria from oxidative stress in vivo and vitro studies [42]. Many studies had confirmed that HA IA injection could reduce pain and stiffness and improve knee function when treating KOA [43,44]. In this review, m-HA injections as the control groups were observed to consistently provide beneficial effects in reducing pain and stiffness and improving knee function during 12-month follow-ups. PRP was isolated from whole blood samples using multiple centrifugations, and it contained various bioactive factors, including VEGF, transforming growth factor-β (TGF-β), PDGF, and bFGF [45,46,47]. The TGF and PDGF could promote cell proliferation and migration during tissue healing, and bFGF plays a vital role in modulating cartilage regeneration [48,49]. These cytokines could inhibit chondrocyte apoptosis, promote chondrocyte proliferation, modulate local inflammation, and reconstruct bone and vessels [50,51]. In addition, other bioactive factors released by PRP contributed to tissue restoration [50]. Because of its potential bioactive function, PRP has received considerable attention in the area of KOA treatments and showed expected effects [52]. In the enrolled studies, PRPs were administered in two to four doses during one to two months, suggesting that m-PRP injections showed more effectiveness with respect to treatments than single PRP injections [53,54]. In this review, m-PRP injections as the experimental groups were found to introduce more effects in lower VAS scores, lower WOMAC scores, and higher KIDC scores in patients with KOA than m-HA injections.

Pain is the main symptom for KOA patients, which severely decreases the quality of life [55] and the function of joints [56]. After long-term disease, patients with KOA showed higher pain sensitivities at the knee joints [57], further lowering treatment sensitivities. Moreover, older patients presented poor prognoses during the treatment [58], affecting the treatment. Therefore, safely, quickly, and effectively relieving pain was the first priority for KOA patients [59]. Currently, the IA HA injection is a popular strategy for treating KOA, which could provide short-term pain relief after injection [60]. m-HA IA injections were effective and safe treatments used in long-term treatments for KOA [61], while in our systematic review, we compared the pain-relieving ability of multiple HA injections and PRP injections, and the results confirmed that m-PRP injections could provide more effective pain relief for KOA patients for up to 6 months. In contrast, this study could not show early-stage clinical effects after injections, especially from 1 to 4 weeks. Most clinical trials did not present clinical effects after the first intervention, and this may be due to local swelling and pain at the puncturing point after injection [22,26], which would influence the effect. Filardo et al. [22,26] showed that PRP injections produced significantly serious post-injection swelling and pain with respect to HA, and this reaction was self-limiting, requiring no medical intervention [22].

In addition, the pain and stiffness of KOA showed a negative effect on walking and movements and reduced the function of joints [62]. Therefore, the impaired function of joints was also an important outcome for evaluating PRP effects. The WOMAC was an osteoarthritis index questionnaire, and it was widely used to assess pain, articular stiffness, and functional limitation [63]. Moreover, the IKDC is a commonly used questionnaire in patients with knee diseases [64]. The results of WOMAC and IKDC in this systemic review presented that multiple IA PRP injections could decrease the WOMAC score and increase the IKDC score, which indicated that the PRP could significantly improve knee functions.

Besides pain relief and function improvement, patients with KOA had a higher quality of life after m-PRP injections at 6 month (WMD = 2.58; 95%CI, −0.37 to 5.52, *p* = 0.09) follow-ups. The EQ-VAS questionnaire shows patients’ self-rated health state by using a visual analog scale [65]. The result presented that both m-PRP and m-HA injections could increase patients’ EQ-VAS scores, while the patients in the m-PRP group had higher scores than m-HA group. The EQ-VAS scores were subjective scores from patients; when patients felt pain-free or less stiffness, they could live and work without symptoms, and they would think they were healthier than before. This result confirmed that m-PRP injections could significantly improve the quality of life, more than m-HA injections.

In all enrolled studies, no patient reported severe complications after the intervention, which proved that both multiple IA PRP injections and multiple IA HA injections were safe. In contrast, the results found that the local swelling and pain of injection sites occurred frequently in the PRP group [22,30,35,36]. This might be because the volume of PRP was larger than the volume of HA in the same study (Table 2). According to enrolled studies, these local complications merely appeared in the early stage of injections; thus, multiple IA PRP injections were available for patients with KOA.

In this study, there were some limitations that need to be noted. The primary limitation was that the doses and intervals of injections were inconsistent in the enrolled RCT. In some RCTs, the intervals of PRP and HA injections were different [27,30,34,37]. The different doses and intervals enhance the bias of outcomes. Secondly, the procedure of PRP was different in the included RCTs, including leucocyte-poor PRP and leucocyte-rich PRP. The different methods may show the influence of leucocytes which may enhance the bias of outcomes. Thirdly, although we tried to maximize the search strategy and enroll as many as studies possible, the number of RCTs was still limited in the study. In addition, the relatively scattered follow-up points in the enrolled led to the fact that not many studies were included in each outcome indicator, which might affect the authenticity of the results.

## 5. Conclusions

For patients with KOA, m-PRP injections could effectively relieve pain, enhance the function of joints, and improve quality of life compared with m-HA injections via the VAS scores, MOWAC scores, IKDC scores, and EQ-VAS scores. Although there are limited reported studies, m-PRP injections are recommended as adjuvant therapies for treating KOA. Furthermore, the PRP preparation, injection intervals, and dosage should be standardized in studies. According to the included RCTs, one or two weeks were suggested as the PRP intervals, and 4-6 ml were suggested as the PRP dosage. Of course, more studies need to be conducted to confirm the best injectional intervals and dosage and compare the effect of LR-PRP and LP-PRP. Besides, large-scale trials with long-term follow-ups need to be conducted in the future to determine the complications of m-PRP injections in treating KOA.

## Figures and Tables

**Figure 1 jpm-13-00429-f001:**
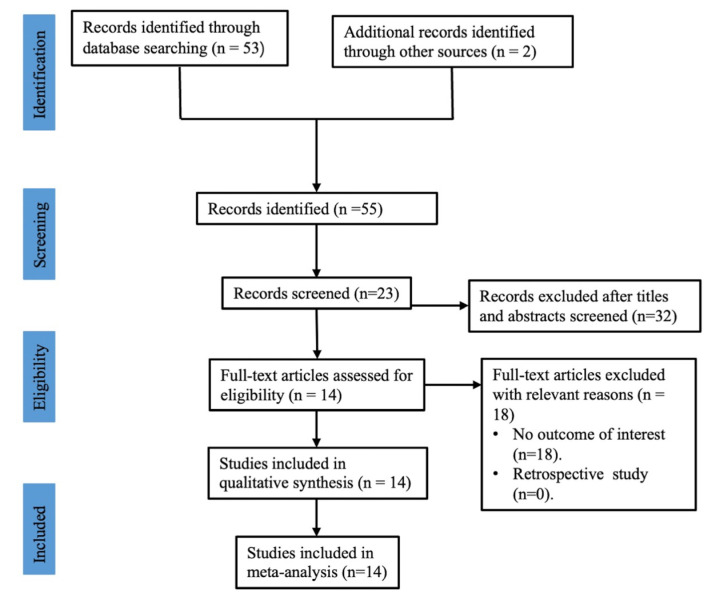
Flow chart of study identification and selection.

**Figure 2 jpm-13-00429-f002:**
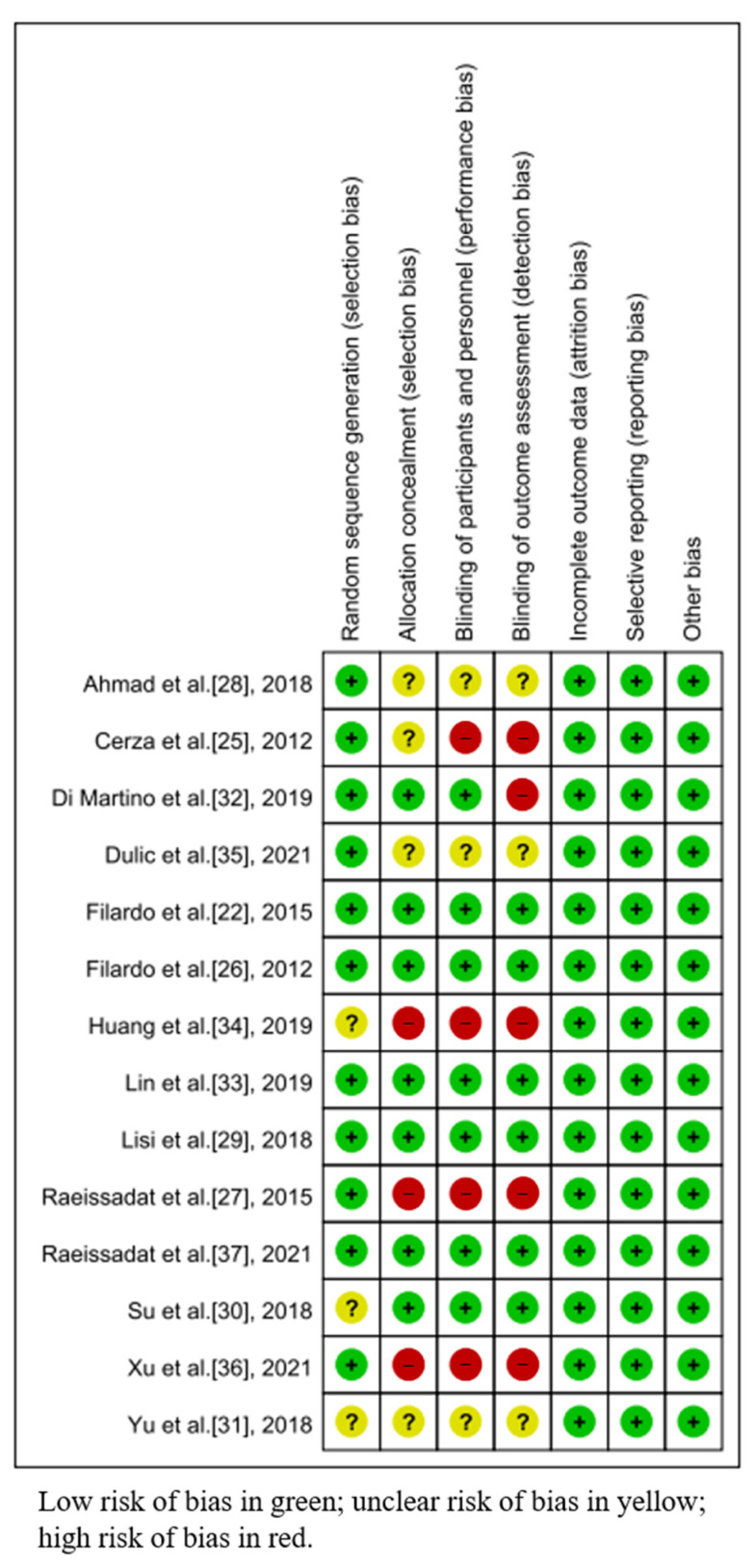
Risk of bias graph.

**Figure 3 jpm-13-00429-f003:**
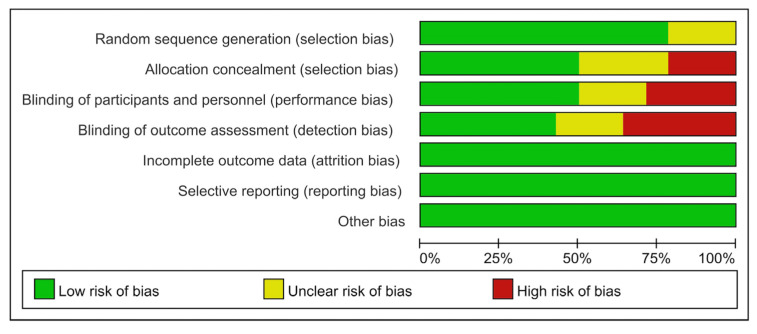
Risk of bias summary: risk of bias is shown as the percentage across all enrolled studies that point out the proportion of different levels of risk of bias for each item.

**Figure 4 jpm-13-00429-f004:**
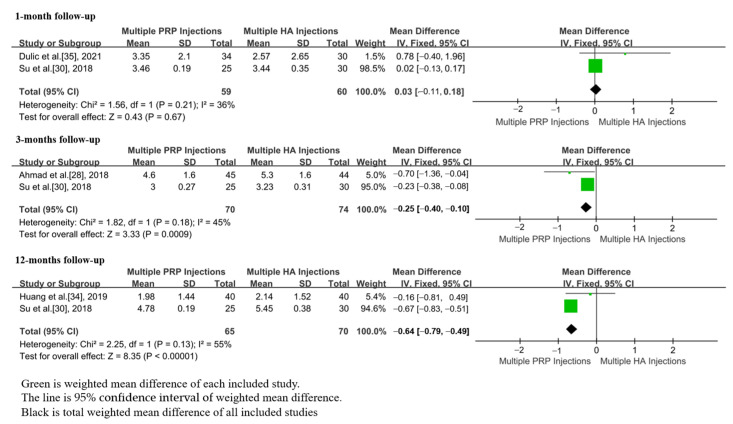
Forest plot showing VAS scores of m-PRP injections versus m-HA injections.

**Figure 5 jpm-13-00429-f005:**
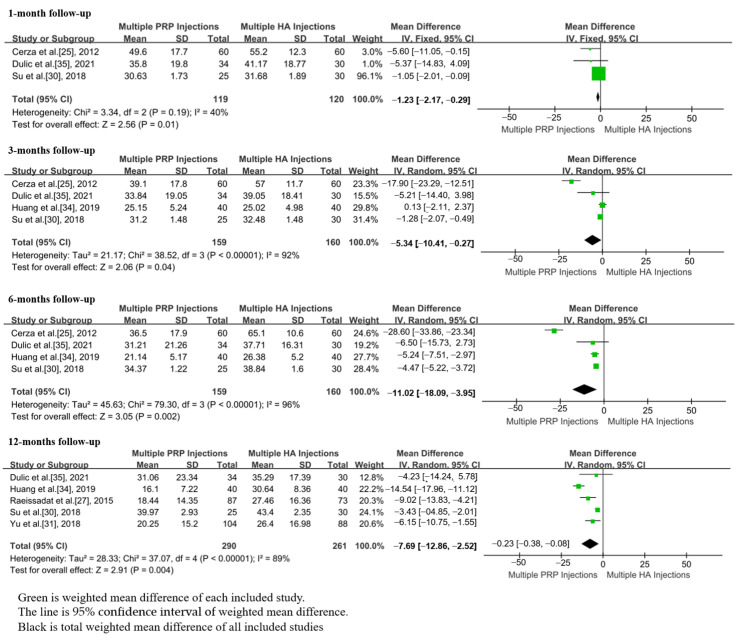
Forest plot showing WOMAC scores of m-PRP injections versus m-HA injections.

**Figure 6 jpm-13-00429-f006:**
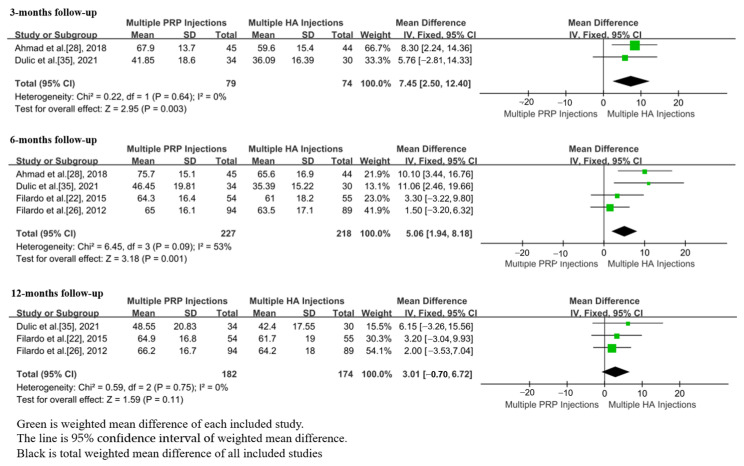
Forest plot showing IKDC scores of m-PRP injections versus m-HA injections.

**Figure 7 jpm-13-00429-f007:**
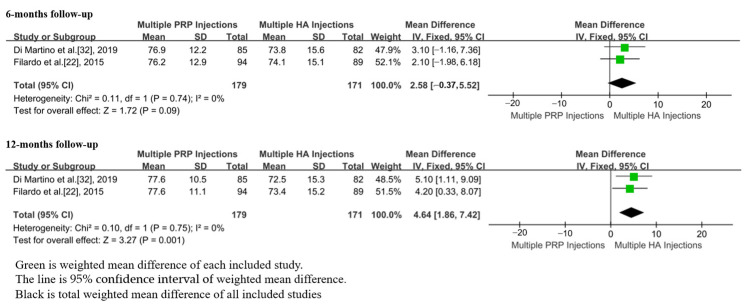
Forest plot showing EQ-VAS scores of m-PRP injections versus m-HA injections.

**Figure 8 jpm-13-00429-f008:**
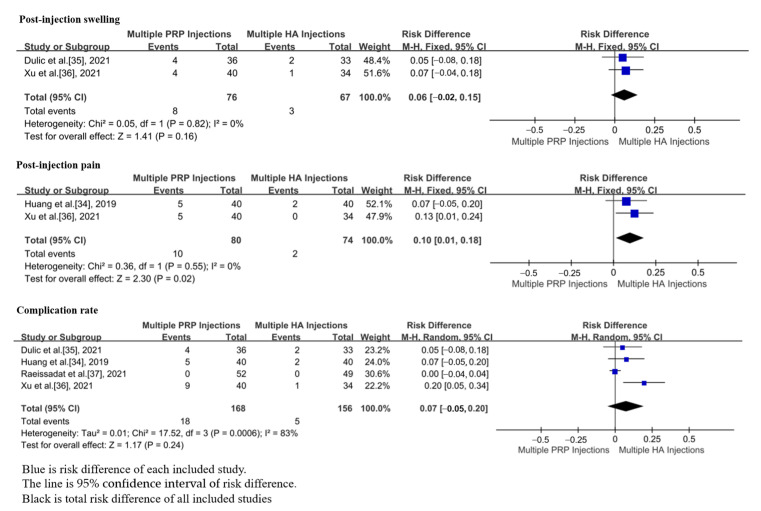
Forest plot showing adverse effects of m-PRP injections versus m-HA injections, including post-injection swelling, post-injection pain, and complication rate.

**Table 1 jpm-13-00429-t001:** The demographic characteristics of all enrolled studies.

Author (Year)	Country	Patients (P/H)	Age (Years) (P/H)	Male (P/H)	Interventions (P/H)	Follow-Up (Months)
Cerza et al. [25], 2012	Rome, Italy	60/60	66.5 (11.3)/66.2 (10.6)	25/28	PRP 4 IA/HA 4 IA	6
Filardo et al. [26], 2012	Bologna, Italy	54/55	55/58	37/31	PRP 3 IA/HA 3 IA	12
Filardo et al. [22], 2015	Bologna, Italy	94/89	53.32 ± 13.2/57.55 ± 11.8	60/52	PRP 3 IA/HA3 IA	12
Raeissadat et al. [27], 2015	Tehran, Iran	87/73	56.85 ± 9.13/61.13 ± 7.48	8/15	PRP 2 IA/HA 3 IA	12
Ahmad et al. [28], 2018	Mansoura, Egypt	45/44	56.2 ± 6.8/56.8 ± 7.4	14/14	PRP 3 IA/HA 3 IA	6
Lisi et al. [29], 2018	Pavia, Italy	30/28	54.4(15.1)/57.1(10.0)	20/16	PRP 3 IA/HA 3 IA	6
Su et al. [30], 2018	Hebei, China	25/30	54.16 ± 6.56/53.13 ± 6.41	11/12	PRP 2 IA/HA 5 IA	18
Yu et al. [31], 2018	Shanxi, China	104/88	46.2 ± 8.6/51.5 ± 9.3	50/48	PRP 4 IA/HA 4 IA	12
Lin et al. [33], 2019	Taiwan	31/29	61.17 ± 13.08/62.53 ± 9.9	9/10	PRP 3 IA/HA3 IA	12
Huang et al. [34], 2019	Jining, China	40/40	54.5 ± 1.2/54.8 ± 1.1	25/19	PRP 3 IA/HA3 IA	12
Di Martino et al. [32], 2019	Bologna, Italy	85/82	52.7 ± 13.2/57.5 ± 11.7	53/47	PRP 3 IA/HA3 IA	24
Dulic et al. [35], 2021	Belgrade, Serbia	34/30	58.8 ± 11.2/59.4 ± 14.0	15/13	PRP 3 IA/HA3 IA	12
Xu et al. [36], 2021	Guiyang, China	40/34	56.9± 4.2/57.1 ± 3.4	10/5	PRP 3 IA/HA3 IA	12
Raeissadat et al. [37], 2021	Tehran, Iran	52/49	56.09 ± 6.0/57.91 ± 6.7	13/12	PRP 2 IA/HA 3 IA	12

**Table 2 jpm-13-00429-t002:** Intervention information for all enrolled studies.

Author (Year)	Injection Frequency	PRP Preparation	Excluding Platelet Count	HA Component	PRP Dosage	HA Dosage
Cerza et al. [25], 2012	Both PRP and HA were performed once a week.	No reported preparation. LP-PRP	Less than 150,000/μL	20 mg/2 mL (Hyalgan, Fidia, Abano Terme, Italy)	5.5 mL	2 mL
Filardo et al. [26], 2012	Both PRP and HA were performed once a week.	150 mL venous blood underwent 2 centrifugations (1480 rpm for 6 min and 3400 rpm for 15 min), producing 20 mL PRP; 5 mL every time; LR-PRP	Less than 150,000/μL	Molecular weight < 1500 kDa, (Hyalubrix, Fidia, Abano Terme (PD), Italy)	5 mL	NR
Filardo et al. [22], 2015	Both PRP and HA were performed once a week.	150 mL venous blood underwent 2 centrifugations (1480 rpm for 6 min and 3400 rpm for 15 min) producing 20 mL PRP; 5 mL every time; LR-PRP	Less than 150,000/μL	Molecular weight < 1500 kDa, (Hyalubrix 30 mg/2 mL, Fidia SpA)	5 mL	2 mL
Raeissadat et al. [27], 2015	PRP was performed once every 4 weeks; HA was performed once a week.	35–40 mL venous blood underwent 2 centrifugations (1600 rpm for 15 min and 2800 rpm for 7 min), producing 4–6 mL PRP. LR-PRP	Less than 150,000/mL	Molecular weight 500,000–730,000 Da. (Hyalgan, Fidia Farmaceutici S.p.A., Abano Terme, Italy)	4–6 mL	2 mL
Ahmad et al. [28], 2018	Both PRP and HA were performed once every 2 weeks.	8 mL venous blood underwent centrifugation (3500 rpm for 9 min) producing 4 mL PRP; LR-PRP	NR	NR	4 mL	2 mL
Lisi et al. [29], 2018	Both PRP and HA were performed once every 4 weeks.	20 mL venous blood underwent centrifugation (900 rpm for 7 min), producing PRP.	NR	20 mg/2 mL (Hyalgan; Fidia, Abano Terme, Italy)	NA	2 mL
Su et al. [30], 2018	PRP was performed once every 2 weeks; HA was performed once a week.	45 mL venous blood underwent 2 centrifugations (1480 rpm for 6 min and 3400 rpm for 15 min) producing 7 mL PRP. LR-PRP	NR	Molecular weight was 0.6–1.5 million Daltons. (Freda, Shandong, China)	6 mL	2 mL
Yu et al. [31], 2018	Both PRP and HA were performed once a week.	PRP from Sigma-Aldrich (Merck KGaA, Darmstadt, Germany)	NR	Only HA (Sigma-Aldrich; Merck KGaA)	2–14 mL	NR
Lin et al. [33], 2019	Both PRP and HA were performed once a week.	10 mL venous blood underwent centrifugation (1500 rpm for 8 min), producing 5 ± 0.5 mL PRP. LP-PRP.	Less than 150,000/μL	Hyruan Plus, 20 mg/2 mL; molecular weight > 2500 kDa; (LG Chem, Seoul, Republic of Korea)	2 mL	2 mL
Huang et al. [34], 2019	PRP was performed once every 3 weeks; HA was performed once a week.	8 mL venous blood underwent centrifugation (3500 rpm for 5 min), producing PRP. LP-PRP.	Less than 150,000/L	Sodium hyaluronate, molecular weight 500–730 kDa (SK chemical research Co., Ltd., Tokyo, Japan)	NR	NR
Di Martino et al. [32], 2019	Both PRP and HA were performed once a week.	150 mL venous blood underwent 2 centrifugations (1480 rpm for 6 min and 3400 rpm for 15 min), producing 20 mL PRP. LR-PRP.	Less than 150,000/μL	Molecular weight > 1500 KDa, 30 mg/2 mL, (Hyalubrix; Fidia SpA).	5 mL	2 mL
Dulic et al. [35], 2021	Both PRP and HA were performed once a week.	60 mL venous blood underwent 2 centrifugations producing PRP. LP-PRP.	NR	Molecular weight 4000 kDa, (Cartinorm, Goodwill Pharma, Hungary)	NR	2 mL
Xu et al. [36], 2021	Both PRP and HA were performed once every 2 weeks.	36 mL venous blood underwent 2 centrifugations (160 G for 10 min and 250 G for 15 min), producing PRP. LP-PRP.	NR	Molecular weight 2500 kDa, (SOFAST, 2 mL/20 mg, Shandong, China)	4 mL	2 mL
Raeissadat et al. [37], 2021	PRP was performed once every 3 weeks; HA was performed once a week.	35 mL venous blood underwent 2 centrifugations (1600 rpm for 15 min and 3500 rpm for 7 min), producing 2 mL PRP. LR-PRP.	Less than 150,000/μL	Molecular weight between 500 to 730 kDa, (Hyalgan, Fidia Farmaceutici S.p.A., Abano Terme, Italy)	2 mL	NR

NR, not reported. Da, Dalton. kDa, kilo Dalton.

**Table 3 jpm-13-00429-t003:** Modified Jadad Score for clinical trials. The score is used to assess the quality of clinical trials; when trials achieved a score of ≥ 4 points, they were considered high quality.

Study (Year)	Randomization	Concealment of Allocation	Double Blinding	Total Withdrawals and Dropouts	Total
Cerza et al. [25], 2012	*	-	-	*	2
Filardo et al. [26], 2012	**	**	**	*	7
Filardo et al. [22], 2015	**	**	**	*	7
Raeissadat et al. [27], 2015	**	-	-	*	3
Ahmad et al. [28], 2018	*	*	*	*	4
Lisi et al. [29], 2018	**	**	**	*	7
Su et al. [30], 2018	**	-	-	*	3
Yu et al. [31], 2018	*	*	*	*	4
Lin et al. [33], 2019	**	**	**	*	7
Huang et al. [34], 2019	*	-	-	*	2
Di Martino et al. [32], 2019	*	*	**	*	5
Dulic et al. [35], 2021	**	-	-	*	3
Xu et al. [36], 2021	*	**	**	*	6
Raeissadat et al. [37], 2021	**	**	**	*	7

*, Each asterisk means one point.

## Data Availability

No new data were created or analyzed in this study. Data sharing is not applicable to this article.

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
