# Peer review of "Multiple Injections of Platelet-Rich Plasma Versus Hyaluronic Acid for Knee Osteoarthritis: A Systematic Review and Meta-Analysis of Current Evidence in Randomized Controlled Trials"

_jpm, 2023, doi:10.3390/jpm13030429_

Round 1

Reviewer 1 Report

Major comments:

The authors have compared the efficacy of PRP injection vs. hyaluronic acid injection and have performed a meta-analysis of existing studies.

The abstract fails to provide the conclusions that answer the question posed (see below).

There are lots of fundamental mistakes in language, and the manuscript urgently needs to be revised by a native speaker.

In the result part it becomes clear that only a very small number of studies is available for each measure that is compared between PRP and HA.

Minor comments:

Abstract:

… a sequence of clinical trials…. This sounds as if every trial was building on the previous one, therefore I recommend to eliminate the term “sequence” in this context.

… remain in disagreement is not the right term. Either the studies remain in disagreement, which basically says that some studies showed something opposite to other studies. Or the clinical benefits remain doubtful / in question / or similar, which means that there is no overall evidence. Please adapt.

… to compare the multiple….. This is redundant with the previous sentence and can be abbreviated.

Studies cannot show “their” postoperative…. Please rephrase.

“Compared with multiple doses of intra-articular HA injections, multiple doses of intra-articular PRP injections can effectively relieve the pain, enhance knee function….” The wording is not clear, as the authors have set out to evaluate the effect of PRP vs. HA. So the authors need to do three things:

1)     Is HA treatment significantly different from doing nothing, and more importantly, is it different from placebo injection (e.g. using saline)

2)     Is PRP treatment significantly different from doing nothing, and more importantly, is it different from placebo injection (e.g. using saline)

3)     Is PRP treatment significantly different (more effective) from HA injections

The 3rd point is the main one, but the answer is not provided in the conclusions. Without that comparison, I think the meta-analysis has failed to provide what it set out for and what should be done.

Introduction:

First sentence: pain, stiffness, swelling, and dysfunction ….. please add of what ?

Line 31: cause that…. Is not proper English

Line 31: The incidence was raised…. is not proper English

Line 34: Is the financial burden on the patients, or on the health care system?

Line 35: clinical (pain / symptomatic) progression, or structural progression. I think the pain progression can be managed, the structural progression cannot.

Line 36: so that pain and stiffness were constantly complained… This is not proper English

Line 37: Even among patients who underwent joint replacement, there were still parts of patients suffering from chronic pain[7]. This is out of context of the current study

Line 40: KOA progression: Please state whether clinical or structural progression

DISCUSSION:

The first few sentences are just a repetition of the introduction

Line 256: Our systematic review was performed to investigate the clinical efficacy of multiple IA PRP injections for KOA.”. How the introduction was set up, the purpose should be testing the hypothesis that PRP is more effective than HA, otherwise the title, as written, makes little sense.

Line 259: “According to this research, multiple PRP injections are more effective in relieving the pain and restoring knee function than multiple HA injections”: If this is so, than there is no need for the current meta-analysis.

Line 265: There is little point in discussion steroid treatment, as this is not what the title says.

Line 273 ff: This part clearly is for the introduction, if at all, but this is not a discussion of the results, which is what this section should focus on.

Line 310. “The result indicates that both 310 multiple PRP injections and multiple HA injections could increase patients’ EQ-VAS 311 scores, while the patients in the PRP group had higher scores.” Please do not say what the results “indicate”, but rather state what they “show”. For a comparison between PRP and HA, you need to state whether the scores were statistically significantly different, and you need to indicate the confidence intervals of both, and the statistical probability of the result being right (or wrong).

Author Response

Thank you for your comments concerning our manuscript entitled “Multiple injections of platelet-rich plasma versus hyaluronic acid for knee osteoarthritis: a systematic review and meta-analysis of current evidence in randomized controlled trials” Thank you very much for your comments on our manuscript! The comments are very valuable and helpful for revising and improving our manuscript. We have carefully handled these comments and made some corrections, hoping to get approval. The corrections and the responses to the reviewer’s comments are as follows:

Comments 1:The abstract fails to provide the conclusions that answer the question posed (see below).

Response and revision 1: Thanks for your kind reviews and comments. We have added related content to the manuscript. Please see the text highlighted in red in the manuscript (Line 12-15, and Line 21-30).

Comments 2: There are lots of fundamental mistakes in language, and the manuscript urgently needs to be revised by a native speaker.

Response and revision 2: Thanks for your kind reviews and comments. The manuscript has been polished by a native speaker.

Comments 3: In the result part it becomes clear that only a very small number of studies is available for each measure that is compared between PRP and HA.

Response and revision 3: Thanks for your kind reviews and comments. This is a limitation of this study, so we added related content to the manuscript. Please see the text highlighted in red in the manuscript (Line 340-347).

Comments 4: … a sequence of clinical trials…. This sounds as if every trial was building on the previous one, therefore I recommend eliminating the term “sequence” in this context.

Response and revision 4: Thanks for your kind reviews and comments. We have revised the manuscript according to the suggestion.

Comments 5: … remain in disagreement is not the right term. Either the studies remain in disagreement, which basically says that some studies showed something opposite to other studies. Or the clinical benefits remain doubtful / in question / or similar, which means that there is no overall evidence. Please adapt.

Response and revision 5: Thanks for your kind reviews and comments. We have revised the manuscript. Please see the text highlighted in red in the manuscript (Line 12-15).

Comments 6: … to compare the multiple….. This is redundant with the previous sentence and can be abbreviated.

Response and revision 6: Thanks for your kind reviews and comments. We have revised the manuscript according to the suggestions.

Comments 7: Studies cannot show “their” postoperative…. Please rephrase.

Response and revision 7: Thanks for your kind reviews and comments. We have revised the manuscript according to the suggestions.

Comments 8: “Compared with multiple doses of intra-articular HA injections, multiple doses of intra-articular PRP injections can effectively relieve the pain, enhance knee function….” The wording is not clear, as the authors have set out to evaluate the effect of PRP vs. HA. So the authors need to do three things:1) Is HA treatment significantly different from doing nothing, and more importantly, is it different from placebo injection (e.g. using saline). 2) Is PRP treatment significantly different from doing nothing, and more importantly, is it different from placebo injection (e.g. using saline). 3) Is PRP treatment significantly different (more effective) from HA injections The 3rd point is the main one, but the answer is not provided in the conclusions. Without that comparison, I think the meta-analysis has failed to provide what it set out for and what should be done.

Response and revision 8: Thanks for your kind reviews and comments. We have revised the manuscript according to the suggestions. Please see the text highlighted in red in the manuscript (Line 21-30).

Comments 9: First sentence: pain, stiffness, swelling, and dysfunction ….. please add of what ?

Response and revision 9: Thanks for your kind reviews and comments. We have revised the manuscript according to the suggestions. Please see the text highlighted in red in the manuscript (Line 36).

Comments 10: Line 31: cause that…. Is not proper English

Response and revision 10: Thanks for your kind reviews and comments. We have revised the manuscript according to the suggestions. Please see the text highlighted in red in the manuscript (Line 38).

Comments 11: Line 31: The incidence was raised…. is not proper English.

Response and revision 11: Thanks for your kind reviews and comments. We have revised the manuscript according to the suggestions. Please see the text highlighted in red in the manuscript (Line 37).

Comments 12: Line 34: Is the financial burden on the patients, or on the health care system?

Response and revision 12: Thanks for your kind reviews and comments. We have revised the manuscript according to the suggestions. Please see the text highlighted in red in the manuscript (Line 40).

Comments 13: Line 35: clinical (pain / symptomatic) progression, or structural progression. I think the pain progression can be managed, the structural progression cannot.

Response and revision 13: Thanks for your kind reviews and comments. We have revised the manuscript according to the suggestions. Please see the text highlighted in red in the manuscript (Line 44).

Comments 14: Line 36: so that pain and stiffness were constantly complained… This is not proper English

Response and revision 14: Thanks for your kind reviews and comments. We have revised the manuscript according to the suggestions. Please see the text highlighted in red in the manuscript (Line 44-45).

Comments 15: Line 37: Even among patients who underwent joint replacement, there were still parts of patients suffering from chronic pain[7]. This is out of context of the current study.

Response and revision 15: Thanks for your kind reviews and comments. We have deleted this sentence according to the suggestions.

Comments 16: Line 40: KOA progression: Please state whether clinical or structural progression.

Response and revision 16: Thanks for your kind reviews and comments. We have revised the manuscript according to the suggestions. Please see the text highlighted in red in the manuscript (Line 45).

Comments 17: The first few sentences are just a repetition of the introduction

Response and revision 3: Thanks for your kind reviews and comments. We have deleted related content in the manuscript.

Comments 18: Line 256: Our systematic review was performed to investigate the clinical efficacy of multiple IA PRP injections for KOA.”. How the introduction was set up, the purpose should be testing the hypothesis that PRP is more effective than HA, otherwise the title, as written, makes little sense.

Response and revision 18: Thanks for your kind reviews and comments. We have added related content in the manuscript. Please see the text highlighted in red in the discussion (Line 258-263).

Comments 19: Line 265: There is little point in the discussion steroid treatment, as this is not what the title says.

Response and revision 19: Thanks for your kind reviews and comments. We have added related content in the manuscript. Please see the text highlighted in red in the discussion (Lines 272-274).

Comments 20: Line 273: This part clearly is for the introduction, if at all, but this is not a discussion of the results, which is what this section should focus on.

Response and revision 20: Thanks for your kind reviews and comments. We have added related content in the manuscript. Please see the text highlighted in red in the discussion (Line 279-285, 290-295).

Comments 21: Line 310. “The result indicates that both 310 multiple PRP injections and multiple HA injections could increase patients’ EQ-VAS 311 scores, while the patients in the PRP group had higher scores.” Please do not say what the results “indicate”, but rather state what they “show”. For a comparison between PRP and HA, you need to state whether the scores were statistically significantly different, and you need to indicate the confidence intervals of both, and the statistical probability of the result being right (or wrong).

Response and revision 21: Thanks for your kind reviews and comments. We have added related content in the manuscript. Please see the text highlighted in red in the discussion (Line 322-323, 326-329).

Once again, thanks to all reviewers for their valuable reviews and comments!

We really hope that the revisions in the manuscript and our accompanying responses will be sufficient to make our manuscript suitable for publication in Journal of Personalized Medicine. If you have any questions, please do not hesitate to contact me.
 Best wishes,

Fengchao Chen
Medical Cosmetic Center, Beijing Friendship Hospital, Capital Medical University, Beijing 100050, China

Reviewer 2 Report

Authors should clearly underline that variability with whole blood starting volumes (12- 150ml!), and separation protocols with PRP will create feasibly enormous inter-patient variability. This is overcome to some extent by the repeat interventions (at very short 1-4 week intervals) for PRP. Typically - we will not see a significant clinical effect in 1-2 weeks (perhaps week three and more likely by week 4). Therefore - in most cases - the PRP was administered on a 2nd, 3rd, or 4th dose at intervals where the authors may not have seen the full effect of the first intervention if the PRP dose was high enough. 

In addition, the authors should explain the following. Excluding platelet count, no differentiation was made as to the PRP WBC characteristics. Was it LR-PRP or LP-PRP? It is becoming increasingly clear that LR-PRP may have a more robust and perhaps longer-lasting effect than LP-PRP. The field’s infatuation that WBC was a bad thing for the joint based on catabolic protein release from in-vitro studies appears unsubstantiated. HA dosing was kept pretty tight across studies, unlike PRP interventions.

It is common in the US (not sure about abroad) for docs to add steroidal to their HA injections to prevent iatrogenic acute inflammatory reactions with HA preparations. This was not differentiated, so I assume it was not involved in the HA arm. If it was - it could be argued that at least some of the HA group effects were due to steroidal implantation as opposed to a pure HA effect - unclear from my read.

Author Response

Thank you for your comments concerning our manuscript entitled “Multiple injections of platelet-rich plasma versus hyaluronic acid for knee osteoarthritis: a systematic review and meta-analysis of current evidence in randomized controlled trials” Thank you very much for your comments on our manuscript! The comments are very valuable and helpful for revising and improving our manuscript. We have carefully handled these comments and made some corrections, hoping to get approval. The corrections and the responses to the reviewer’s comments are as follows:

Comments 1: Authors should clearly underline that variability with whole blood starting volumes (12- 150ml!), and separation protocols with PRP will create feasibly enormous inter-patient variability. This is overcome to some extent by the repeat interventions (at very short 1-4 week intervals) for PRP. Typically - we will not see a significant clinical effect in 1-2 weeks (perhaps week three and more likely by week 4). Therefore - in most cases - the PRP was administered on a 2nd, 3rd, or 4th dose at intervals where the authors may not have seen the full effect of the first intervention if the PRP dose was high enough.

Response and revision 1: Thanks for your kind reviews and comments. We added related content in the manuscript. Please see the text highlighted in red in the manuscript (Line 170-176, 320-326)

Comments 2: In addition, the authors should explain the following. Excluding platelet count, no differentiation was made as to the PRP WBC characteristics. Was it LR-PRP or LP-PRP? It is becoming increasingly clear that LR-PRP may have a more robust and perhaps longer-lasting effect than LP-PRP. The field’s infatuation that WBC was a bad thing for the joint based on catabolic protein release from in-vitro studies appears unsubstantiated. HA dosing was kept pretty tight across studies, unlike PRP interventions.

Response and revision 2: Thanks for your kind reviews and comments. We added related content in the manuscript. Please see the text highlighted in red in the manuscript (Table 2, Line 354-361)

Comments 3: It is common in the US (not sure about abroad) for docs to add steroidal to their HA injections to prevent iatrogenic acute inflammatory reactions with HA preparations. This was not differentiated, so I assume it was not involved in the HA arm. If it was - it could be argued that at least some of the HA group effects were due to steroidal implantation as opposed to a pure HA effect - unclear from my read.

Response and revision 3: Thanks for your kind reviews and comments. We added related content in the manuscript. Please see the text highlighted in red in the manuscript (Table 2, Line 181-182)

Once again, thanks to all reviewers for their valuable reviews and comments!

We really hope that the revisions in the manuscript and our accompanying responses will be sufficient to make our manuscript suitable for publication in Journal of Personalized Medicine. If you have any questions, please do not hesitate to contact me.
 Best wishes,

Fengchao Chen
Medical Cosmetic Center, Beijing Friendship Hospital, Capital Medical University, Beijing 100050, China

Reviewer 3 Report

Level of English is insufficient and needs editing by a native English speaker for example; proving that multiple injections have better effect- line 60

 Aim clearly stated.

How were unpublished papers found and defined?

Latin such as in vivo should be in italics

What is meant by To maximize the search, we also used a method of backward chaining of refer-85 ences from retrieved papers ?

Table 1- abbreviations should be defined under the table.

Author Response

Thank you for your comments concerning our manuscript entitled “Multiple injections of platelet-rich plasma versus hyaluronic acid for knee osteoarthritis: a systematic review and meta-analysis of current evidence in randomized controlled trials” Thank you very much for your comments on our manuscript! The comments are very valuable and helpful for revising and improving our manuscript. We have carefully handled these comments and made some corrections, hoping to get approval. The corrections and the responses to the reviewer’s comments are as follows:

Comments 1: Level of English is insufficient and needs editing by a native English speaker for example; proving that multiple injections have a better effect- line 60

Response and revision 1: Thanks for your kind reviews and comments. The manuscript has been polished by a native speaker.

Comments 2: Aim clearly stated.

Response and revision 2: Thanks for your kind reviews and comments.

Comments 3: How were unpublished papers found and defined?

Response and revision 3: Thanks for your kind reviews and comments. We added related content to the manuscript. Please see the text highlighted in red in the manuscript (Line 97)

Comments 4: Latin such as in vivo should be in italics

Response and revision 4: Thanks for your kind reviews and comments. We have revised the manuscript according to the suggestions.

Comments 5: What is meant by To maximize the search, we also used a method of backward chaining of refer-85 ences from retrieved papers?

Response and revision 5: Thanks for your kind reviews and comments. We added related content to the manuscript. Please see the text highlighted in red in the manuscript (Line 90-91).

Comments 6: Table 1- abbreviations should be defined under the table.

Response and revision 6: Thanks for your kind reviews and comments. We added related content to the manuscript. Please see the text highlighted in red in the manuscript (Table 1)

Once again, thanks to all reviewers for their valuable reviews and comments!

We really hope that the revisions in the manuscript and our accompanying responses will be sufficient to make our manuscript suitable for publication in Journal of Personalized Medicine. If you have any questions, please do not hesitate to contact me.
 Best wishes,

Fengchao Chen
Medical Cosmetic Center, Beijing Friendship Hospital, Capital Medical University, Beijing 100050, China

Round 2

Reviewer 1 Report

The authors have undertaken some changes in the manuscript. Yet, to refer the reviewer to the MS to see the changes is insufficient. PLease structure the response in:

Reviewer comment

Author comment

Author action.
For the latter, please paste all changes made to the manuscript pertaining to this point directly below your author comment (and say where these can be found in the manuscript), so that the reviewer can go through the review letter without having to jump back and forth between the letter and the manuscript, as this is quite tedious.

Author Response

We have uploaded the point-by-point response in the attachment. Please see the attachment.

Reviewer 2 Report

The authors made a signifficant efforts to improve the paper quality. However, I still have several comments:

1. Description of Figures 4,5,6,7 (Forest plots) should be more detailed. Legend (describing symbols in green, black, blue) is missing,

2. I would like authors to discuss clinical efficacy of PRP vs HA related to mechanism of action.

Author Response

(The authors gave the same response as above.)
